# Screening for Novel Inhibitors of Amyloid Beta Aggregation and Toxicity as Potential Drugs for Alzheimer’s Disease

**DOI:** 10.3390/ijms241411326

**Published:** 2023-07-11

**Authors:** Sanaa Sharari, Nishant N. Vaikath, Magdalini Tsakou, Simona S. Ghanem, Kostas Vekrellis

**Affiliations:** 1Neurological Disorder Research Center, Qatar Biomedical Research Institute (QBRI), Hamad Bin Khalifa University (HBKU), Qatar Foundation, Doha P.O. Box 5825, Qatar; sanaa_sharai_92@hotmail.com (S.S.); mtsakou@hbku.edu.qa (M.T.); sghanem@hbku.edu.qa (S.S.G.); 2Center of Basic Research, Biomedical Research Foundation of the Academy of Athens, 11527 Athens, Greece; kvekrellis@yahoo.gr

**Keywords:** Alzheimer’s disease, amyloid beta, aggregation, amyloid fibrils, salvianolic acid, ginsenoside Rb1, dihydromyricetin, treatment discovery

## Abstract

AD is the most common neurodegenerative disorder characterized by progressive memory impairment and cognitive deficits. The pathology of AD is still unclear; however, several studies have shown that the aggregation of the Aβ peptide in the CNS is an exclusively pathological process involved in AD. Currently, there is no proven medication to cure or prevent the disease progression. Nevertheless, various therapeutic approaches for AD show only relief of symptoms and mostly work on cognitive recovery. However, one of the promising approaches for therapeutic intervention is to use inhibitors for blocking the Aβ peptide aggregation process. Recently, herbal phenolic compounds have been shown to have a therapeutic property for treatment of AD due to their multifaceted action. In this study, we investigated the effectiveness of SA, Gn Rb1, and DMyr on inhibiting the aggregation and toxicity of Aβ40 and Aβ42 using different biochemical and cell-based assays. Our results showed that SA and DMyr inhibit Aβ40 and Aβ42 fibrillation, seeded aggregation, and toxicity. Gn Rb1 did not have any effect on the aggregation or toxicity induced by Aβ40 and Aβ42. Moreover, SA and DMyr were able to disaggregate the preformed fibrils. Overall, these compounds may be used alone or synergistically and could be considered as a lead for designing new compounds that could be used as effective treatment of AD and related disorders.

## 1. Introduction

Alzheimer’s disease (AD) is a progressive neurodegenerative disorder caused by gradual loss of cholinergic neurons in limbic and neocortical regions [1]. AD is characterized by the presence of neurofibrillary tangles and neuritic plaques, commonly referred to as senile plaques. Neurofibrillary tangles primarily consist of tau aggregates. Abnormal tau proteins un-dergo hyperphosphorylation, resulting in their misfolding and subsequent aggregation into insoluble tangles within neurons. These tangles disrupt normal brain cell function, impair neuronal communication, and ultimately lead to neurodegeneration [1]. The plaques primarily consist of 39- to 43-residue Amyloid beta (Aβ) peptides, which are derived from the transmembrane amyloid-beta precursor protein (APP) through endoproteolytic cleavage. APP are cleaved by gamma secretase, resulting in the formation of Aβ fragments. The accumulation of Aβ as plaques leads to neuronal death (Figure 1) [2]. This neurodegeneration leads to a depletion in glutamate and acetylcholine neurotransmitters, which correlates with the cognitive impairment and behavioral changes that AD patients experience [3]. Neuropathological studies have shown the buildup of amyloid beta (Aβ) plaque deposits outside of and neurofibrillary tangles in neurons, causing a decrease in cholinergic neurotransmission and cognitive impairment [4]. Currently, the focus of research is the amyloid hypothesis of AD, which states that the accumulation of Aβ outside of neurons is responsible for cognitive impairment and neuronal death [5]. This accumulation triggers an inflammatory response, synaptic dysfunction, and alters neuronal homeostasis, leading to neuronal death and AD [6,7]. Studies have supported the cause of AD due to misfolded aggregates of human Aβ42, rather than more abundant Aβ40 [8,9].

In its native form, Aβ plays a normal physiological role in healthy individuals. However, in AD patients, Aβ expression increases, leading to the formation of aggregates, which are β-sheet structures that are deposited as senile plaques [10,11,12]. Similar to any other amyloid protein and peptides, Aβ undergoes a highly dynamic self-assembly process into amyloid fibrils, resulting in the formation of various intermediates with differences in size, structure, and shape. Recent studies show that the aggregation process proceeds in a nucleation-dependent manner, forming mature fibrils through intermediate stages such as oligomers and protofibrils. Increasing evidence suggests that these prefibrillar soluble oligomers rather than the mature fibrils of Aβ are responsible for neurodegeneration and synaptic dysfunction in AD. Moreover, mature fibrils can indirectly contribute to neuronal damage by binding to and activating microglia [6,13]. Current drugs for AD only provide symptomatic relief and do not prevent or reverse the disease. Therefore, searching for compounds that can inhibit the formation of early Aβ aggregates may be a promising strategy for therapeutic intervention in AD and related disorders [4,14].

Plant-derived natural polyphenols, such as those found in tea, nuts, berries, and cocoa, have been found to have a demonstrated promising effect on the aggregation of Aβ peptides in vitro, owing to their aromatic phenolic structure [15]. Studies have highlighted the neuroprotective properties of specific compounds such as myricetin and ginsenoside Rb1 (GnRb1), which exhibit inhibitory effects on the progression of AD [16,17,18,19]. Furthermore, salvianolic acid B (SA), through its antioxidative and anti-inflammatory effects, has been shown to possess neuroprotective function in an Aβ25–35 peptide-induced mouse model of AD [20]. In addition, Tang et al. demonstrated that SA inhibits Aβ generation by modulating APP processing in SH-SY5Y-APPsw cells [21].

Given the common process of aggregation among amyloid protein, drugs targeting the aggregation process of one disease may exhibit promising effects on others. Previous studies on α-synuclein, a protein implicated in Parkinson’s disease that shares a similar aggregation pattern with Aβ peptides, have shown that Gn Rb1 acts as a potent inhibitor of α-synuclein aggregation and toxicity of [22].

The formation of Aβ plaques resulting from accumulation of Aβ in the brain is widely recognized as the primary pathological hallmark of AD. Consequently, targeting the aggregation process of both Aβ40 and Aβ42 peptides holds considerable promise as a therapeutic strategy for managing the disease. In this study, we investigated the effect of Gn Rb1, SA, and DMyr on the aggregation of Aβ40 and Aβ42, with the aim of evaluating their potential as drug candidates for AD treatment.

## 2. Results

### 2.1. SA and DMyr, but Not Gn Rb1, Inhibit the Aggregation of Aβ40 and Aβ42 In Vitro

Aβ40 and Aβ42 solutions (100 µM) either alone or with SA, Gn Rb1, and DMyr at different molar ratios (Aβ: compounds; 1:1, 1:5, and 1:10) were incubated at 37 °C for 14 days. The aggregation was monitored by the Th-S fluorescence assay at regular time intervals.

We found that both SA and DMyr inhibit the aggregation of Aβ40 and Aβ42 in a concentration-dependent manner (Figure 2). For Aβ42, the inhibition of aggregation was observed as early as 2 days of incubation with DMyr (Figure 2F), and when used at a molar excess of 1:10, a complete inhibition was observed at day 4 and day 10 for DMyr and SA, respectively (Figure 2B,F). In contrast, the inhibition effect of SA and DMyr on Aβ40 was most prominent after 10 days of incubation, and a complete inhibition of aggregation was observed in both compounds at ratio 1:10 (Figure 2A,E). However, using Gn Rb1, we did not observe any significant effect on the process of aggregation for either Aβ40 or Aβ42 (Figure 2C,D).

These findings were further confirmed by electron microscopy. TEM images showed that in the presence of SA or DMyr, Aβ40 and Aβ42 formed less dense, thin, and short fibrils, and small early aggregates, in a concentration-dependent manner, as compared to dense meshes of long fibrils formed by aged Aβ40 and Aβ42 alone (Figure 3A,B,E,F). In the presence of Gn Rb1, TEM images were similar to Aβ40 and Aβ42 alone (Figure 3C,D), consistent with the Th-S fluorescence assay.

Next, we measured the levels of Aβ40 and Aβ42 early aggregates formed during the aggregation process. Aliquots of Aβ40 and Aβ42 incubated alone or with different ratios of compounds (Aβ: compounds; 1:1, 1:5, and 1:10) during the aggregation assay were taken and assessed by oligomeric ELISA using Aβ-specific antibody 6 × 10^10^. We observed that with the addition of either SA or DMyr, stabilizes the fibrillation, as observed by the constant signal obtained during aggregation; whereas in the control samples, the level of Aβ40 and Aβ42 signal decreased after day 6, depicting the formation of mature fibrils containing a smaller number of exposed epitopes (Figure 4). However, in the Aβ42 sample containing SA at a 1:10 molar ratio, the signal was less than the control (Figure 4B). Thus, these data demonstrate that SA and DMyr could stabilize the Aβ40 and Aβ2 early aggregation and SA might further inhibit the early aggregation at a high concentration.

### 2.2. SA and DMyr Disaggregate Fibrils of Aβ40 and Aβ42

Due to their high effectiveness in inhibiting Aβ40 and Aβ42 fibrillation, SA and DMyr were also tested for their efficacy in reversing fibrillation. Herein, 25 µM of aged Aβ40 and Aβ42 were incubated at 37 °C alone or with compounds (SA, Gn Rb1, and DMyr) at different molar ratios (Aβ: compounds; 1:5 and 1:10) for 24 h. Assessment of fibril content of the samples at different time intervals was done by the Th-S assay.

Addition of micromolar concentrations of SA or DMyr to the existing fibrils of Aβ40 and Aβ42 led to their disaggregation in a concentration-dependent manner, as determined by a decrease in Th-S signals (Figure 5A,B,E,F). This effect was obvious after 3 h of incubation for both Aβ40 and Aβ42 fibrils, with the Th-S signals showing approximately 1/5th of the control samples (Aβ40 and Aβ42 fibrils alone). The control samples without SA or DMyr did not show any decrease in the Th-S signal.

The third tested compound, Gn Rb1, had no significant effect on the disaggregation of Aβ40 or Aβ42 fibrils given the comparable Th-S signals observed with the control sample at different ratios (Figure 5C,D).

### 2.3. SA and DMyr Block the Seeded Aggregation of Aβ40 and Aβ42 Monomers

It has been reported that the process of aggregation follows a nucleation-dependent pathway [23]. Short fibrils or seeds have been described to speed up the nucleation phase of the aggregation process in vitro and in vivo via a process known as seeding [23]. Hence, we investigated whether the compounds could inhibit the seeded aggregation of Aβ40 and Aβ42. Briefly, Aβ40 and Aβ42 fibrils were fragmented by sonication to short fibrils, which were used as ‘seeds’. These seeds were then added to Aβ40 or Aβ42 monomers and the samples were incubated at 37 °C for 6 h. In the control samples, the additions of seeds accelerated the aggregation of both Aβ40 and Aβ42 monomers, as observed by the increase in the Th-S signal (Figure 6). Interestingly, the addition of SA or DMyr was found to block the seed-induced aggregation of Aβ40 and Aβ42 in a concentration-dependent manner (Figure 6A,B,E,F). Moreover, there was > 90% inhibition of seeded aggregation observed when compounds were used at a 1:10 molar excess concentration of SA or DMyr (Figure 6A,B,E,F). However, Gn Rb1 had no significant effect on the process of seeded aggregation of both peptides given the comparable Th-S signals with control at different ratios (Figure 6C,D).

### 2.4. SA and DMyr Inhibit the Aβ40- and Aβ42-Induced Cytotoxicity in SH-SY5Y Cells

To determine whether SA, DMyr, or Gn Rb1 could block Aβ40- and Aβ42-induced cytotoxicity, we used WT SH-SY5Y, human neuroblastoma cells. Fibrils of Aβ40 or Aβ42 solutions (100 μM) incubated either alone or in the presence of SA, Gn Rb1, and DMyr at different ratios (Aβ: compounds; 1:1 and 1:10) were added to WT SH-SY5Y cells and the viability of treated cells was determined after 4.5 h by the MTT assay.

We found that Aβ40 and Aβ42 fibrils incubated with SA or DMyr were less toxic to the cells in a concentration-dependent manner, as shown by an increase in MTT reduction (Figure 7). Interestingly, at a 1:10 molar excess of SA or DMyr, the viability of the cells was increased by approximately 90% in both peptides (Figure 5). However, Gn Rb1 did not show any cytoprotective effect on SH-SY5Y cells against the Aβ40 or Aβ42 fibril-induced cytotoxicity (Figure 7).

## 3. Discussion

AD is a neurodegenerative disorder characterized by the accumulation of Aβ peptides and neurofibrillary tangles, leading to neuronal loss and cognitive impairment [24]. The precise mechanism through which Aβ leads to cell degeneration remains unclear; however, it is hypothesized that disturbances in Ca2^+^ homeostasis and the induction of oxidative free radical damage contribute to cell death [25]. This amyloid hypothesis is relevant not only to AD but also to other diseases such as Parkinson’s disease [6]. Despite extensive research, there is currently no effective treatment for AD, with existing drugs only providing symptomatic relief [26]. Numerous studies suggest that targeting Aβ aggregation with small molecules may offer a potential therapeutic approach, but a comprehensive understanding of the amyloid structure is crucial for the design of improved molecules for treatment [27,28]. A recent study has reported the promising activity of polyphenol molecules found in plants, such as tea, nuts, berries, and cocoa, in targeting Aβ fibrillation in vitro [15]. Based on the previous studies targeting α-synuclein, a protein implicated in PD and which has a similar aggregation process to Aβ peptides, we aimed to investigate whether GnRb1 could inhibit Aβ aggregation and could be considered as a promising approach in the treatment of AD. In this study, we evaluated the effect of three compounds: SA, which has been studied before [29,30,31,32]; DMyr, which is the derivative of the previously studied compound myricetin [33]; and GnRb1, which has not been studied for Aβ aggregation yet. To achieve our objective, we employed biochemical assays to confirm the previously established hypothesis that the Aβ40 and Aβ42 amyloid hypothesis is mediated by a nucleation-dependent process (Figure 2 and Figure 6). Our results showed that monomeric Aβ42 aggregated more rapidly than Aβ40, and the presence of seeds accelerated the aggregation process of both peptides (Figure 2 and Figure 6). Furthermore, our findings revealed a complex relationship between the formation of early aggregates and fibrils (Figure 4). These results provide support for the nucleation-dependent process underlying Aβ fibril formation, which has been established by previous studies [10,34]. Therefore, exploring the aggregation inhibition effects of SA, Gn Rb1, or DMyr based on the amyloid hypothesis may present a promising therapeutic strategy for AD. Ginseng and its bioactive molecules, such as ginsenoside, have been reported to exhibit a neuroprotective effect through their multiple mechanisms of actions on AD pathology. A recent study demonstrated that Gn Rb1 could improve cognition in AD rats by altering the cleavage process of APP into non-amyloidogenic proteins [35]. However, the effect of Gn Rb1 on amyloid formation has not been assessed yet [36]. In our study, we found that Gn Rb1 does not impact the aggregation process or cell toxicity of Aβ40 and Aβ42. It failed to inhibit the fibrillation process (Figure 2 and Figure 3), seeded fibrillation process (Figure 6), and cell toxicity (Figure 7). Additionally, Gn Rb1 was unable to disaggregate preformed fibrils (Figure 5). Therefore, our results indicate that Gn Rb1 does not inhibit the amyloid formation or toxicity of Aβ40 and Aβ42 in AD, in contrast to its inhibitory effect on α-syn in PD [22]. In contrast, our findings demonstrate that SA and DMyr exhibited strong inhibitory effects on amyloid formation and toxicity of Aβ40 and Aβ42. DMyr specifically inhibited fibrillation, seeded aggregation, and cell toxicity in a concentration-dependent manner. These results are consistent with previous studies that highlight DMyr as a potent inhibitor of Aβ fibrillation and toxicity [37,38,39]. Furthermore, we observed that DMyr was capable of disaggregating preformed fibrils in a concentration-dependent manner (Figure 5E,F). Our results on the inhibitory effect of DMyr and its ability to disaggregate the preformed fibrils are consistent with work on myricetin and DMyr [33,38]. However, our results on the DMyr ability to stabilize formed off-pathway early aggregates (Figure 4C,D) differ with other studies on myricetin and DMyr, which showed that it can block the formation of oligomers [38,40]. Furthermore, investigation with SA has revealed that its inhibitory effect on the fibrillation and toxicity of Aβ42 stems from the stabilization of formed off-pathway early aggregates, which aligns with our findings (Figure 2, Figure 3, Figure 4 and Figure 7) [29]. Moreover, our study showed high concentrations of SA may inhibit the early aggregation of Aβ42 (Figure 4). These results are consistent with recent work that report the concentration-dependent inhibition of Aβ40 fibrillation by SA and its ability to disaggregate the Aβ40 fibrils (Figure 2, Figure 3 and Figure 5) [30]. Additionally, our findings showed that SA could prevent the seeded fibrillation of Aβ40 and Aβ42 (Figure 6A,B). In conclusion, among the three compounds investigated, SA and DMyr exhibited inhibitory effects on the aggregation and toxicity of Aβ40 and Aβ42 by stabilizing off-pathway early aggregates (Table 1). However, Gn Rb1 failed to inhibit the aggregation process of Aβ40 and Aβ42 (Table 1). Based on our in vitro findings, further investigation using an in vivo model is warranted to assess the ability of these compounds to remove plaques and inhibit seeded aggregation of Aβ40 and Aβ42. SA and DMyr represent a potential starting point for the development of novel compounds that could be utilized as drugs for the treatment of AD and related disorders.

## 4. Methods and Materials

### 4.1. Aggregation of Aβ40 and Aβ42 In Vitro

Aβ40 and Aβ42 (purity > 90–95%) peptides were received from Dr. James I. Elliott at Yale University (New Haven, CT, USA) and The ERI Amyloid Laboratory (Oxford, CT, USA). SA, Gn Rb1, and DMyr were purchased from the National Institute for the Control of Pharmaceutical and Biological Products, China (NICPBP). Fresh monomers of Aβ40 and Aβ42 solutions were prepared by dissolving the peptide with a minimum amount of DMSO, followed by adding it drop by drop into the required volume of sterile PBS such that the final DMSO in the solution was not more than 5%. The solution was then ultracentrifuged at 50,000 rpm for 1 h at 4 °C. The top 75% of the supernatant was taken into sterile tubes and this solution was considered as the fresh monomers. In total, 200 µM of Aβ40 and Aβ42 monomers was incubated either alone or in the presence of an equal volume of SA, Gn Rb1, or DMyr at the different molar ratios of 1:1, 1:5, and 1:10 (Aβ40/42: compounds) in a total volume of 800 μL. The tubes were sealed with paraffin and incubated at 37 °C for 14 days. The aggregation of Aβ40 and Aβ42 solutions in the presence or absence of tested compounds was monitored by a Th-S assay by taking aliquots at day 0, 2, 4, 6, 8, 10, 12, and 14.

### 4.2. Thioflavin-S (Th-S) Assay

Aβ40 and Aβ42 fibril formation was assessed by the Th-S binding assay. Th-S is a fluorescent dye that binds with fibrils containing β-sheet structures. For this assay, 10 µL of samples was loaded in triplicate in a 384-well black plate (Nunc), then diluted with 40 µL of the Th-S solution to give a final concentration of 5 μM of the sample and 20 μM of the Th-S solution. Fluorescence was then measured by a PerkinElmer automated plate reader, with the excitation and emission wavelengths set at 430 and 550 nm, respectively. The fluorescence of the blank was subtracted from all sample readings.

### 4.3. Aβ40 and Aβ42 Disaggregation Assay

Aβ40 and Aβ42 solutions (50 μM, in PBS) were aggregated as described above for 14 days. Aβ40 and Aβ42 aggregates were then incubated with equal volume of PBS or in the presence of compounds (SA, Gn Rb1, or DMyr) at various molar ratios (Aβ: compound ratio; 1:5 and 1:10) to give a final concentration of Aβ40 and Aβ42 of 25 μM. The samples were incubated at 37 °C for 24 h. Aliquots were collected at regular time intervals and the Th-S assay was performed immediately.

### 4.4. Seeding Polymerization Assay

The aggregation of fresh monomers Aβ40 and Aβ42 with or without seeds was tested as described previously [22]. Briefly, monomeric Aβ40 and Aβ42 peptide solutions (50 µM) were seeded with 1 µM of seeds and incubated with or without SA, Gn Rb1, or DMyr at different molar ratios (monomer: compound; 1:1, 1:5, and 1:10) at 37 °C for 6 h. The aggregation was monitored by the Th-S assay as described before.

### 4.5. Transmission Electron Microscopy (TEM)

The samples (5 µL) were deposited on carbon-coated 400-mesh copper grids (TED PELLA INC), fixed briefly with 0.5% glutaraldehyde (5 µL), negatively stained with 2% uranyl acetate (5 µL), and then examined with a Talos-200c transmission electron microscope (FEI, Thermofisher, Waltham, MA, USA).

### 4.6. ELISA for Measuring Aβ40 and Aβ42 Early Aggregates

A 384-well clear Maxisorb plate (Nunc) was coated with 1 µg/mL (50 µL/well) of a non-biotinylated 6 × 10^10^ antibody (Aβ-specific antibody) diluted in 200 mM NaHCO_3_, pH 9.6, and incubated overnight at 4 °C. Then, the plate was blocked with 100 µL/well of a blocking buffer (PBS containing 2.25% gelatin and 0.5% tween) for 2 h at 37 °C. After washing with PBST (PBS containing 0.05% Tween-20), 50 µL/well of the samples were added and incubated at 37 °C for 2 h. Then, 50 µL/well of the biotinylated 6 × 10^10^ (1 µg/mL) antibody diluted in the blocking buffer was added and incubated at 37 °C for 1 h. After washing 4 times with PBST, 50 µL/well of extravidin–peroxidase diluted to 1:7500 in the blocking buffer was added and incubated at 37 °C for 1 h. A TMB substrate was added, incubated for 20 min, and then 50 µL/well of a stop solution (0.6 N H_2_SO_4_) was added. The absorbance at 450 nm was measured using the Perkin Elmer plate reader.

### 4.7. Culture of SHSY-5Y Human Neuroblastoma Cell Lines

The WT SHSY-5Y human neuroblastoma cell line was cultured in DMEM/F-12 (1:1) (Hyclone-cat number: SH30271.01) containing 15% FBS, 1% P/S, and 1% MEM NEAA (Gibco ref No: 11140-035). Then, cells were kept at 37 °C in a humidified incubator with 5% CO_2_/95% air.

### 4.8. Measurement of Cell Viability

SHSY-5Y cells suspended in DMEM/F-12 were plated at a density of 15,000 cells in a 96-well plate. After 24 h, the medium was replaced with 200 µL of OPTI-MEM 1x (Gibco-USA) reduced serum media containing an aged solution with or without SA, Gn Rb1, and DMyr at different ratios (1:1 and 1:10). Then, cells were incubated for 48 h in 5% CO_2_ at 37 °C. A total of 20 µL of MTT (3-(4, 5-dimethylthiazol-2-yl)-2,5-diphenyltetrazolium bromide) (Sigma-Aldrich, St. Louis, MO, USA) (6 mg/mL) in PBS was dispensed into each well, and the plate was incubated at 37 °C for 4.5 h. The MTT-containing medium was removed, and replaced with 100 µL/well of a lysis buffer (15% SDS, 50% N,N-dimethylformamide, pH 4.7) overnight at 37 °C. The absorbance at 590 nm was measured using the Perkin Elmer plate reader (Perkin Elmer, Waltham, MA, USA).

## Figures and Tables

**Figure 1 ijms-24-11326-f001:**
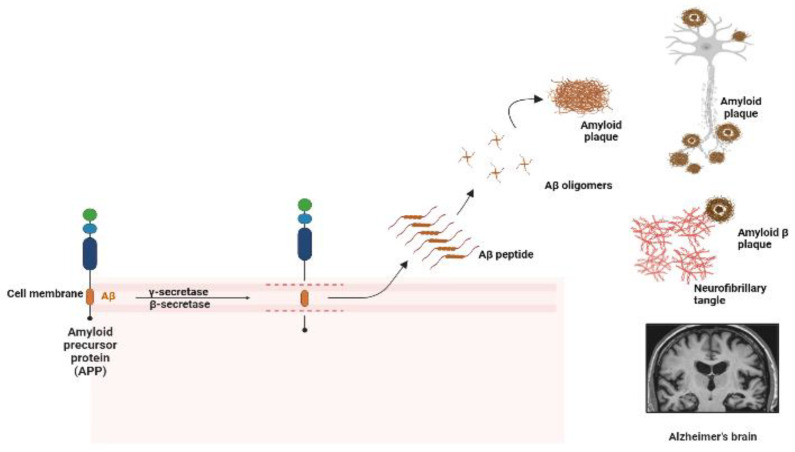
The cascade of amyloid hypothesis in AD. Toxic peptides start to accumulate to form oligomers, which trigger a cascade of events, leading to amyloid plaque deposition and neurological symptoms of AD.

**Figure 2 ijms-24-11326-f002:**
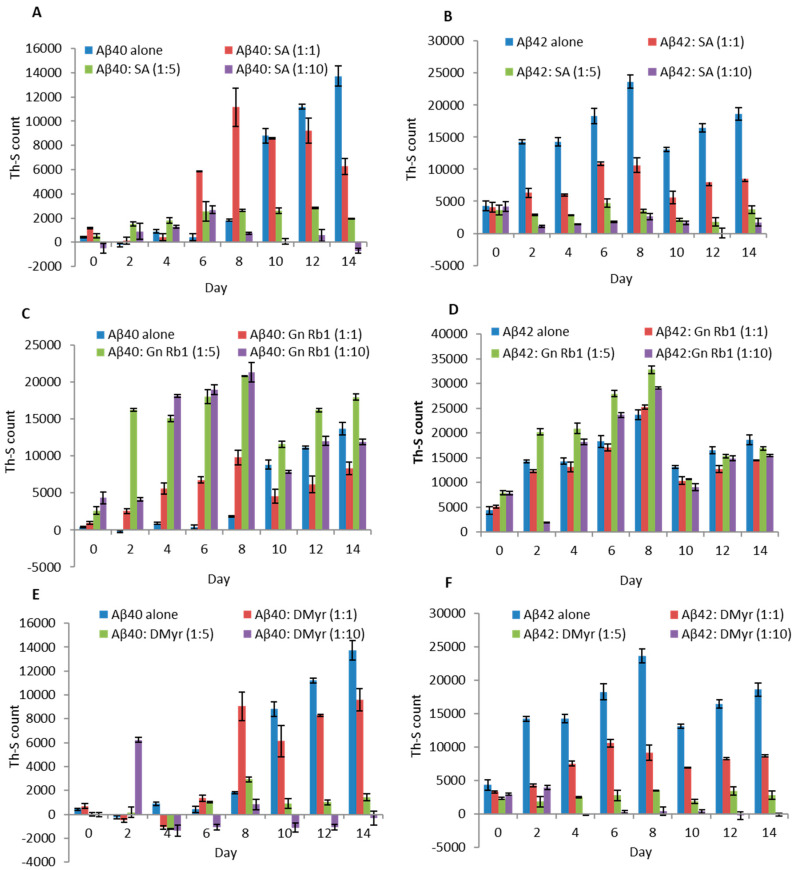
Effect of Gn Rb1, SA, and DMyr on aggregation of Aβ40 and Aβ42. Aβ40 (**A**,**C**,**E**) and Aβ42 (**B**,**D**,**F**) solutions at 100 µM were incubated at 37 °C for 14 days alone or with compounds SA (**A**,**B**), Gn Rb1 (**C**,**D**), and DMyr (**E**,**F**) at different ratios (Aβ: compounds; 1:1, 1:5, and 1:10). Readings were taken in triplicate, and the means ± standard deviations were plotted.

**Figure 3 ijms-24-11326-f003:**
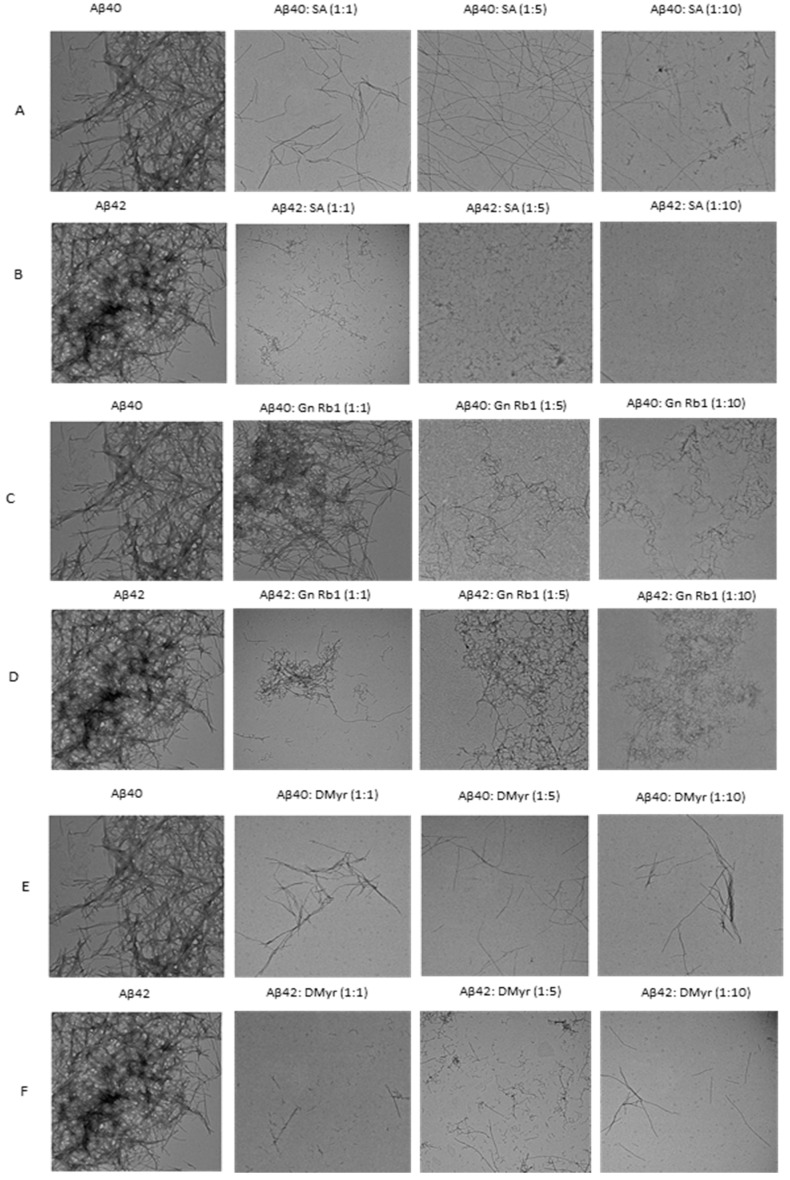
Transmission electron microscopy (TEM) images for aged Aβ40 (**A**,**C**,**E**) and Aβ42 (**B**,**D,F**). Samples incubated alone or in the presence of compounds SA, Gn Rb1, and DMyr at different molar ratios (Aβ: compounds; 1:1, 1:5, and 1:10) for 14 days at 37 °C. Scale bar, 200 nm.

**Figure 4 ijms-24-11326-f004:**
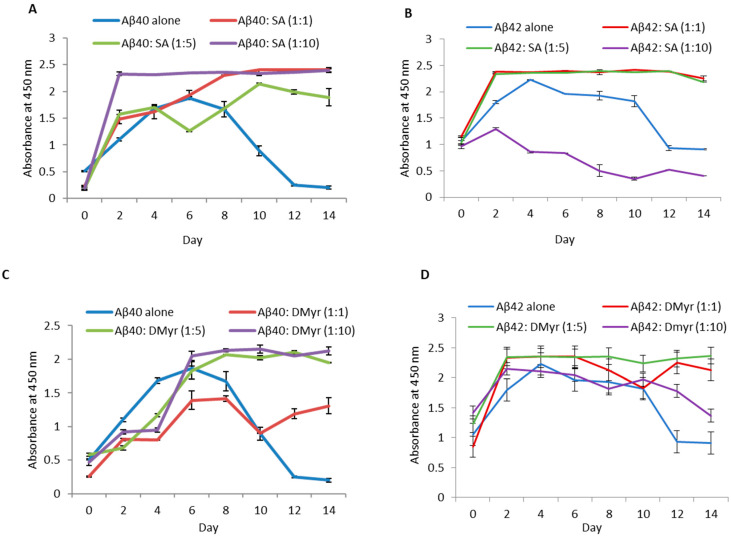
SA and DMyr stabilize the early aggregation of Aβ40 and Aβ42. Aliquots of Aβ40 (**A**,**C**) and Aβ42 (**B**,**D**) at 100 µM incubated alone or with different molar ratios (Aβ: compounds; 1:1, 1:5, and 1:10) at 37 °C for 14 days were tested. ELISA was used to investigate the inhibition effect of SA and DMyr on the oearly aggregation of Aβ. Readings were taken in triplicate, and the means ± standard deviations are shown.

**Figure 5 ijms-24-11326-f005:**
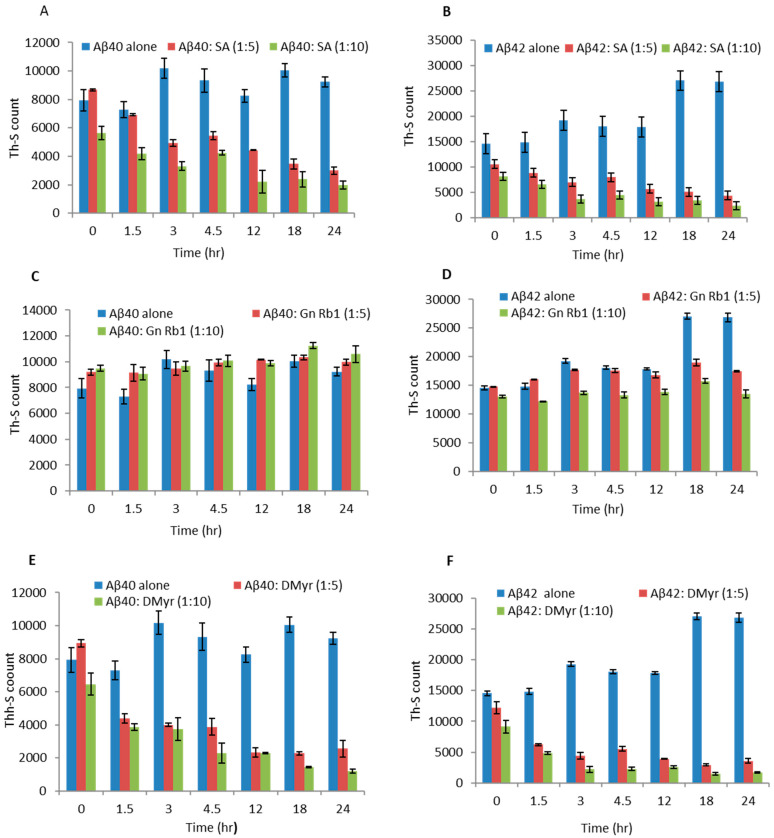
Effect of SA, Gn Rb1, and DMyr on preformed Aβ40 and Aβ42 fibrils. Aged Aβ40 (**A**,**C,E**) and Aβ42 (**B**,**D**,**F**) at 25 µM were incubated alone or in the presence of compounds SA, Gn Rb1, and DMyr at different molar ratios (Aβ: compounds; 1:5 and 1:10) at 37 °C for 24 h. Th-S signal was used to measure the fibril content of the samples. Readings were taken in triplicates, and the means ± standard deviations are shown.

**Figure 6 ijms-24-11326-f006:**
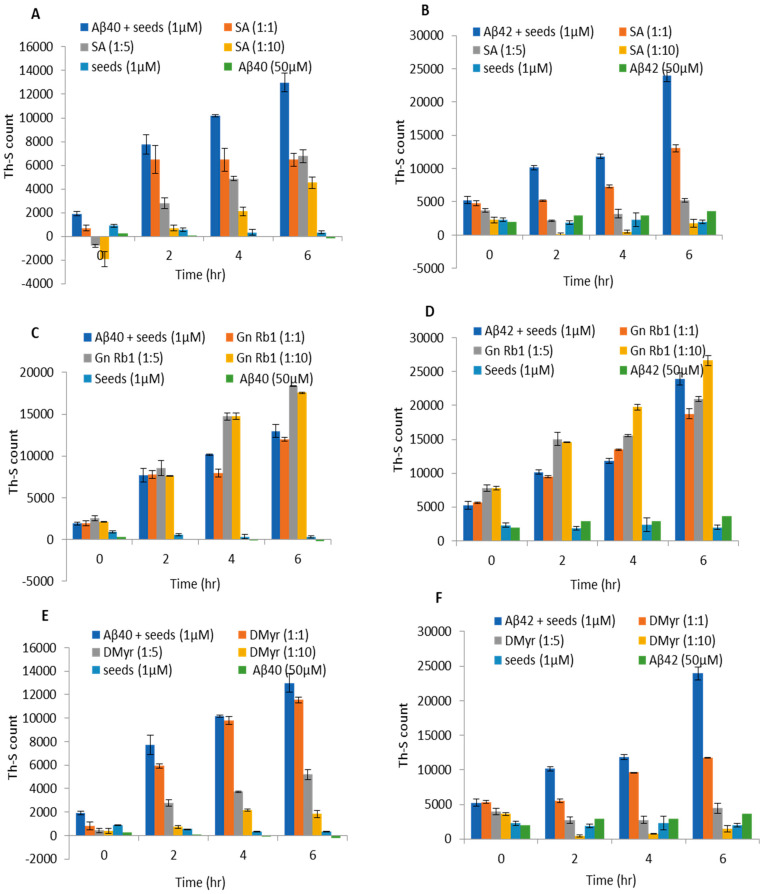
The effect of SA, Gn Rb1, and DMyr on the seeded polymerization of Aβ40 and Aβ42. Solutions of Aβ40 (**A**,**C**,**E**) and Aβ42 (**B**,**D**,**F**) at 50μM were seeded with short fibrils of Aβ40 and Aβ42, respectively, (1 µM) and incubated alone or with SA, Gn Rb1, and DMyr at different molar ratios (Aβ: compounds; 1:1, 1:5, and 1:10) at 37 °C for 6 h. Th-S assay was used to investigate the inhibition effect of compounds on the seeded fibrillation process. Readings were taken in triplicate, and the means ± standard deviations are shown.

**Figure 7 ijms-24-11326-f007:**
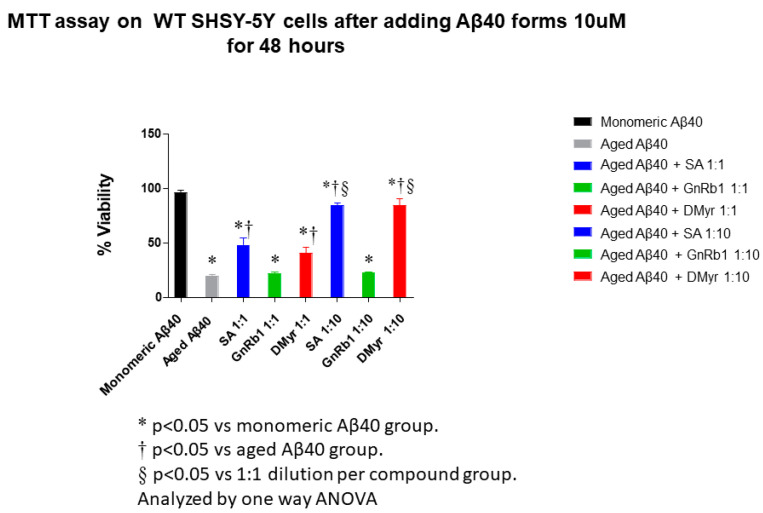
The effect of SA, Gn Rb1, and DMyr on the neuronal cell toxicity triggered by the aggregates of Aβ40 and Aβ42. The viability of SHSY-5Y human cells was evaluated by MTT assay. The results are presented as percentages of the average of the control (i.e., untreated cells). The cells were treated with either Aβ40 or Aβ42 aged with or without SA, Gn Rb1, and DMyr for 48 h prior to the addition of MTT. Graphs A and B illustrate the toxicity of the compounds on Aβ40- and Aβ42-treated cells, respectively (average of 3 wells ± standard deviation). Statistical analysis was performed using one-way ANOVA.

**Table 1 ijms-24-11326-t001:** Summary of findings of the chemical compounds.

Compounds	Abbreviation	Previous Studies	New Findings
**Salvianolic acid B**	**SA**	SA has a neuroprotective function owing to its antioxidative and anti-inflammatory effects [20]. SA performs inhibition activity on Aβ40 fibrillation in a concentration-dependent manner and can disaggregate the Aβ40 preformed fibrils [30]. SA inhibition activity on the fibrillation process and toxicity of Aβ42 is due to stabilization of formed off-pathway early aggregates [29].	SA has a strong inhibition effect on amyloid formation and toxicity of Aβ40 and Aβ42. High concentrations of SA may inhibit the formation of Aβ42 early aggregates.
**Dihydromyricetin**	**DMyr**	DMyr improves the symptoms of AD by overexpression of NEP, which speeds up the Aβ decomposition, in addition to inhibition of inflammatory, oxidative stress, and other non-amyloidogenic pathways [41] DMyr is a strong inhibitor of the Aβ fibrillation process and toxicity [37,38,39]. DMyr inhibits the Aβ40 fibrillation, disaggregates the preformed fibrils, and inhibits the cytotoxicity [38]	DMyr has a strong inhibition effect on amyloid formation and toxicity of Aβ40 and Aβ42. DMyr ability to stabilize formed off-pathway early aggregates is not in agreement with other work on myricetin and DMyr, which showed it can block the formation of early aggregation [40].
**Ginsenoside Rb1**	**Gn Rb1**	Gn Rb1 is a strong inhibitor of the aggregation and toxicity of α-synuclein [22]. Gn Rb1 could improve cognition in AD rats by altering the cleavage process of APP into non-amyloidogenic proteins [35]. The effect of Gn Rb1 on the process of amyloid formation has not been assessed yet [36].	Gn Rb1 failed to inhibit the amyloid formation and toxicity of Aβ40 and Aβ42 in AD as opposed to the inhibition effect on amyloid formation of α-syn in PD [22].

## Data Availability

Not applicable.

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
