# Peer review of "Screening for Novel Inhibitors of Amyloid Beta Aggregation and Toxicity as Potential Drugs for Alzheimer’s Disease"

_ijms, 2023, doi:10.3390/ijms241411326_

Round 1

Reviewer 1 Report

This munuscript deals with inhibitory effects of some compounds to prevent amyloid beta aggregation related to Alzheimer's disease. 

<Major points>

1. The authors confuse Introduction with Discussion. Considerable contents mentioned in Discussion should be addressed in Introduction. In Discussion, the authors do not state sufficient discussion including the interpretation of their experimental results. The authors just repeat the results in Discussion. Please discuss based on your results.

2. Please state the criteria with which the authors selected SA, DMyr, GnRb1as initial drug candidates, among many known potential druggable compounds related to AD.

<Minor points>

1. The authors should explain sufficiently why Ab42 and Ab40 can be targets for inhibiors in this study, although they just simply mentioned about it in Introduction.

2. In Figure 1, simple figures corresponding to the respective steps can help display the contents more effectively.

3. The authors can prepare a table summarizing information on the chemical compounds.

4. Abbreviation usage is incorrect. Abbreviation can be defined when a word is on the first appearance, and the abbreviation can be used once it is mentioned before.

 Moderate editing of English language is necessary.

Author Response

Reviewer One:

Comments and Suggestions for Authors

This manuscript deals with inhibitory effects of some compounds to prevent amyloid beta aggregation related to Alzheimer's disease.

<Major points>

  1. The authors confuse Introduction with Discussion. Considerable contents mentioned in Discussion should be addressed in Introduction. In Discussion, the authors do not state sufficient discussion including the interpretation of their experimental results. The authors just repeat the results in Discussion. Please discuss based on your results.

Author reply: We thank the reviewer for his/her valuable suggestion. We have now modified the introduction and added contents from discussion to the introduction section (page no. 2 line no. 38-46). The discussion part has been now edited and incorporated with inclusion of previous studies (page no. 18 and 19  line no. 348-386).

  1. Please state the criteria with which the authors selected SA, DMyr, GnRb1as initial drug candidates, among many known potential druggable compounds related to AD.

Author reply: We thank the reviewer for the comment. We have now stated the criteria for selection of the compounds SA, DMyr, GnRb1 in the introduction. We have now added the following text in the introduction “Previous studies on α-synuclein, a protein implicated in Parkinson’s disease that shares similar aggregation pattern with Aβ peptides, have shown that Gn Rb1 act as a potent inhibitor of α-synuclein aggregation and toxicity of [22]” (Page no. 4 Line no. 94-97). We used two previously studied compounds (SA and DMyr) as positive controls, and we aimed to clearly understand the mechanisms of action of these two compounds on Aβ40 and Aβ42 aggregation and toxicity. We ended the introduction with the aim of the study “The formation of Aβ plaques resulting from accumulation of Aβ in the brain is widely recognized as the primary pathological hallmark of AD. Consequently, targeting the aggregation process of both Aβ40 and Aβ42 peptides holds considerable promise as a therapeutic strategy for managing the disease. In this study, we investigated the effect of Gn Rb1, SA and DMyr on the aggregation of Aβ40 and Aβ42, with the aim of evaluating their potential as drug candidates for AD treatment” (Page no. 4 Line no. 101-108).

<Minor points>

  1. The authors should explain sufficiently why Ab42 and Ab40 can be targets for inhibiors in this study, although they just simply mentioned about it in Introduction.

Author reply: We thank the reviewer for his/her input. We have elaborated further about the main pathology of AD in introduction part. We added ‘AD is characterized by the presence of neurofibrillary tangles and neuritic plaques, commonly referred to as senile plaques. These plaques primarily consist of 39- to 43- residue Amyloid beta (Aβ) peptides, which are derived from the transmembrane amyloid-beta precursor protein (APP) through endoproteolytic cleavage.  APP are cleaved by gamma secretase, resulting in the formation of Aβ fragments. The accumulation of Aβ as plaques leads to neuronal death [2] ’’ (page no .2  line no.38-46). In addition, we added “Studies have supported the cause of AD due to misfolded aggregates of human Aβ42, rather than more abundant Aβ40 [8], [9]” (page no .2 line no.56-57).We also modified the text in the second paragraph of introduction as “Like any other amyloid protein and peptides, Aβ undergoes a highly dynamic self-assembly process into amyloid fibrils, resulting in the formation of various intermediates with differences in size, structure, and shape. Recent studies show that the aggregation process proceeds in a nucleation-dependent manner, forming mature fibrils through intermediate stages such as oligomers and protofibrils. Increasing evidence suggests that these prefibrillar soluble oligomers rather than the mature fibrils of Aβ are responsible for neurodegeneration and synaptic dysfunction in AD. Moreover, mature fibrils can indirectly contribute to neuronal damage by binding to and activating microglia [6, 13] " (page. No. 3 line no.61-72).

  1. In Figure 1, simple figures corresponding to the respective steps can help display the contents more effectively.

Author reply: We thank the reviewer for his/her suggestion. We have now replaced the text with figure (Figure 1).

  1. The authors can prepare a table summarizing information on the chemical compounds.

Author reply: We appreciate the reviewer’s input. We have now included a table (Table 1) summarizing the information related to each compound.

  1. Abbreviation usage is incorrect. Abbreviation can be defined when a word is on the first appearance, and the abbreviation can be used once it is mentioned before.

Author reply: We have now corrected the usage of abbreviation.

Reviewer 2 Report

The subject of this paper deserves consideration, yet the manuscript is disseminated with typos, mispells and formatting errors that require extensive revision (see overlapping pictures in figure 2B and distorted graph in figure 5 as examples). Without these improvements any review would be really hard to be performed.

I would also refine figure 1 (in the present format it looks like a draft more than a final scheme).

Automated numbering of lines overlapping figures should also be avoided.

Should be improved. 

Author Response

Reviewer Two:

The subject of this paper deserves consideration, yet the manuscript is disseminated with typos, misspells and formatting errors that require extensive revision (see overlapping pictures in figure 2B and distorted graph in figure 5 as examples). Without these improvements any review would be really hard to be performed.

Author Reply: We thank the reviewer for the comment. We have now fixed the typo and grammar with tracked changes. All figures are now thoroughly check and corrected. The overlapping pictures in Figure 2B has been corrected. The graph in Figure 5 also have been rectified.

I would also refine figure 1 (in the present format it looks like a draft more than a final scheme).

Author Reply: We thank the reviewer for his/her suggestion. We have now replaced the text with figure (Figure 1).

Automated numbering of lines overlapping figures should also be avoided.

Author reply: We thank the reviewer for his/her suggestion. We have now corrected the overlapping figures.

Round 2

Reviewer 1 Report

The manuscript has been quite improved.

Minor editing of English language required